# Tolerability of daily intermittent or continuous short-arm centrifugation during 60-day 6° head down bed rest (AGBRESA study)

Timo Frett[1]*, David Andrew Green[2,3,4], Edwin Mulder[1], Alexandra Noppe[1], Michael Arz[1], Willi Pustowalow[1], Guido Petrat[1], Uwe Tegtbur[5], Jens Jordan[1,6]

1 Institute of Aerospace Medicine, German Aerospace Center, Cologne, Germany, 2 Space Medicine Team, European Astronaut Centre, European Space Agency, Cologne, Germany, 3 KBRwyle GmbH, Cologne, Germany, 4 King's College London, London, United Kingdom, 5 Institutes of Sports Medicine, Hannover Medical School, Hannover, Germany, 6 Chair of Aerospace Medicine, University of Cologne, Cologne, Germany

* Timo.frett@dlr.de

**Data Availability Statement:** All relevant data are within the manuscript and its Supporting Information files.

## Abstract

Artificial gravity through short-arm centrifugation has potential as a multi-system countermeasure for deconditioning and cranial fluid shifts that may underlie ocular issues in microgravity. However, the optimal short-arm centrifugation protocol that is effective whilst remaining tolerable has yet to be determined. Given that exposure to centrifugation is associated with presyncope and syncope and in addition motion sickness an intermittent protocol has been suggested to be more tolerable. Therefore, we assessed cardiovascular loading and subjective tolerability of daily short arm centrifugation with either an intermittent or a continuous protocol during long-term head-down bed rest as model for microgravity exposure in a mixed sex cohort. During the Artificial Gravity Bed Rest with European Space Agency (AGBRESA) 60 day 6° head down tilt bed rest study we compared the tolerability of daily +1 $Gz$ exposure at the center of mass centrifugation, either performed continuously for 30 minutes, or intermittedly (6 x 5 minutes). Heart rate and blood pressure were assessed daily during centrifugation along with post motion sickness scoring and rate of perceived exertion. During bed rest, 16 subjects (6 women, 10 men), underwent 960 centrifuge runs in total. Ten centrifuge runs had to be terminated prematurely, 8 continuous runs and 2 intermittent runs, mostly due to pre-syncopal symptoms and not motion sickness. All subjects were, however, able to resume centrifuge training on subsequent days. We conclude that both continuous and intermittent short-arm centrifugation protocols providing artificial gravity equivalent to +1 $Gz$ at the center of mass is tolerable in terms of cardiovascular loading and motion sickness during long-term head down tilt bed rest. However, intermittent centrifugation appears marginally better tolerated, albeit differences appear minor.

**Funding:** The "Artificial Bed Rest study with European Space Agency" (ABRESA) study was financed by NASA and ESA. Our experiments were included into the study conduction and financed by the DLR Institute of Aerospace Medicine. KBRwyle GmbH provided support in the form of salaries only for author D.G, but did not have any additional role in the study design, data collection and analysis, decision to publish, or preparation of the manuscript. The specific roles of these authors are articulated in the 'author contributions' section.

**Competing interests:** The authors declare no competing interests as KBRwyle GmbH had no role in the study design and thus this does not alter our adherence to PLOS ONE policies on sharing data and materials.

## Introduction

Long term space missions elicit multi-system deconditioning including reduced skeletal muscle strength [1], bone mineral density [2], and central blood volume [3–5]. Moreover, sustained cephalad fluid shifts appear to negatively affect ocular health and brain structure, leading to the so-called space associated neuro-ocular syndrome [6, 7]. Furthermore, returning astronauts may experience reduced aerobic capacity [8], and pre-syncopal symptoms indicative of poorer orthostatic tolerance [9]. In an attempt to counter deconditioning on the International Space Station, integrated resistance and aerobic training is prescribed using a number of dedicated devices [10].

Crewmembers train 6–7 days per week with 6–7 resistance and 4–7 cardiovascular sessions per week [11, 12]. Daily training requires approximately 2.5 hours per crewmember including the training, rest periods, and equipment setup, stowage and cleaning. Despite the substantial investment of time, resource and effort this approach is not entirely effective in mitigating musculoskeletal [13], nor aerobic [5, 14] deconditioning, hence physical rehabilitation is required following return to Earth. Moreover, no effective countermeasures against space associated neuro-ocular syndrome currently exist. Thus more effective and ideally more efficient countermeasures are required for future missions to the Moon, and beyond [15].

Artificial gravity through axial acceleration generated by short-arm human provides musculoskeletal loading via the generation of ground reaction forces and an orthostatic challenge through a hydrostatic pressure gradient both of which are absent in microgravity. Indeed, short-arm human centrifugation may attenuate bone, muscle, and cardiovascular deconditioning [16] induced by 6° head-down bed rest. For instance, in short term (5 day) bed rest, an established terrestrial model of cephalad fluid shifts and space-associated deconditioning [17]) studies, daily 30 minutes centrifugation with at least 1 g at the center of mass resulted in no change in postural muscle strength with good tolerability [18, 19]. Furthermore, exposure to artificial gravity appeared to provide protection against post-bed rest orthostatic intolerance [20, 21]. The primary objective of the AGBRESA bed rest study is to compare the protective effects of one single daily bout (30 min) versus multiple daily bouts of AG (6 x 5 min) on physiological functions that are affected by simulated weightlessness during 60 days of bed rest.

However, exposure to exaggerated hydrostatic pressure gravitational gradients induced by centrifugation can elicit presyncopal symptoms or syncope [20, 22, 23]. Furthermore, head movements within a rotating environment are associated with motion sickness symptoms [24]. Yet, to be acceptable as an integrative spaceflight countermeasure, a form of repeated exposure to artificial gravity needs to be tolerable over a long duration mission for both males and females as a number of recent studies have observed sex differences in autonomic cardiovascular control during exposure to orthostatic stress [22, 25–27].

Thus, the aim of our study was to assess the tolerability of daily 30 minute intermittent, or continuous short-arm centrifugation with 1 g at center of mass during 60 days (6°) head-down bedrest in a mixed sex cohort.

## Methods

### Study subjects

This study is part of the NASA/ESA/DLR 60-day 6° head down bed rest study 'Artificial Gravity Bed Rest with European Space Agency' (AGBRESA) that was conducted from March until December 2019 at the: envihab facility of the Institute of Aerospace Medicine of the German Aerospace Center (DLR) in Cologne, Germany. The study enrolled 24 healthy individuals (16 men, 8 women), who had been submitted to detailed medical and psychological screening

having provided written informed consent. The study was approved by the North Rhine Medical Association (2018143 vote from 17.08.2018).

## Protocol

Following a 14-day baseline data collection period, study subjects entered 60 days of strict 6° head-down bed rest. At the end of the baseline data collection phase, participants were pseudo-randomly distributed into 3 groups: a control group with no centrifugation, an intermittent centrifugation group, and a continuous centrifugation group. The intermittent centrifugation group underwent daily 6x5 minutes centrifugation with 3 minutes breaks between runs (Fig 1: Left Panel). The continuous centrifugation group underwent a single daily 30 minute centrifugation run (Fig 1: Right Panel).

All centrifugation was performed using the: envihab short-arm human centrifuge with participants exposed to *+1 Gz* at their center of mass (CoM) and thus approximately *+2 Gz* at foot level. Rotational speed of the centrifuge was calculated individually based upon each subject's anthropometry to determine center of mass (ratio center of mass to body height 56% for male/54% for female). During ramp up/down phases, (de)acceleration did not exceed $5° \, s^{-2}$ to reduce the risk of vestibular-induced tumbling sensations. All subjects underwent two centrifuge familiarization sessions prior to bed rest at the same *+Gz* level as the main study with an intermittent profile of two 5 minute periods separated by a 3 minute break.

Head restrainers were not provided, but participants were instructed to keep their body and head still throughout the centrifugation as much as possible. To assist in maintaining consciousness and limit pre-syncopal symptoms, subjects were trained, prior to the bed rest campaign, in the performance of voluntary isometric calf muscle pump contractions along with (the trunk and gluteal muscles) to promote venous return [28]. However, subjects were instructed to contract only when experiencing significant (pre-syncopal) symptoms such as dizziness or blurred vision.

## Cardiovascular monitoring

During centrifugation, heart rate was continuously recorded via a five lead electrocardiogram in addition to periodic brachial blood pressure (Philips IntelliVue® MP2). In the intermittent

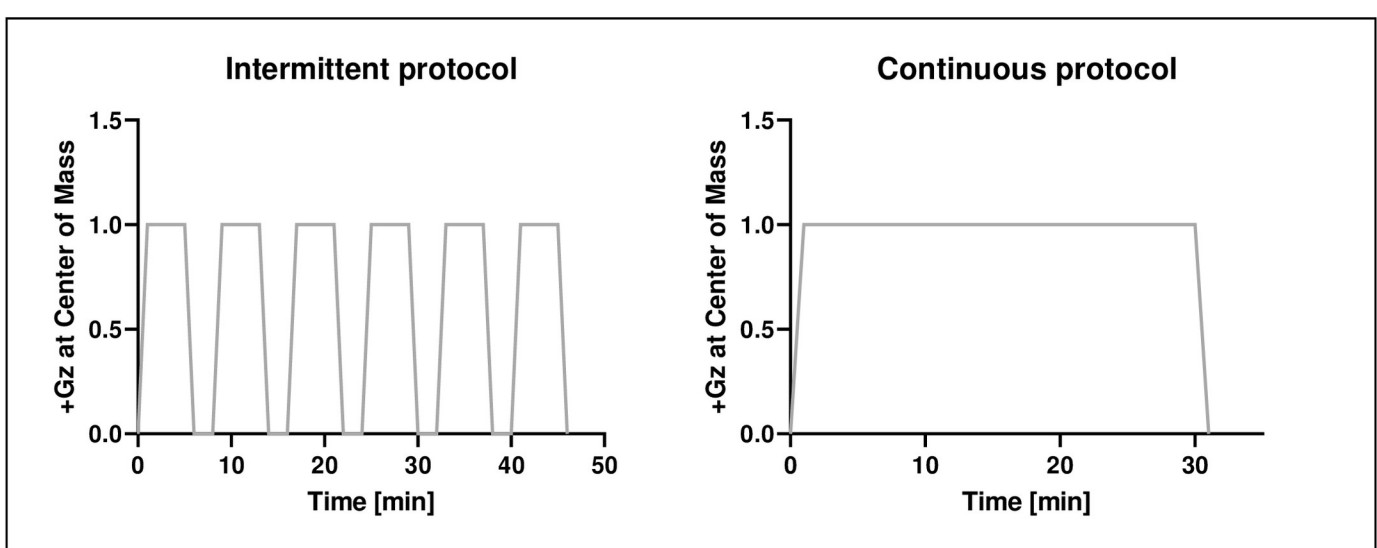

**Fig 1. Artificial gravity was generated by centrifugation with +1Gz at center of mass and approx. +2Gz at feet.** Participants were randomly assigned to an intermittent centrifugation group with 6 x 5 min centrifugation with 3 minute breaks (left side) and a continuous group with 30 min centrifugation (right side).

centrifugation group, blood pressure was recorded 2 minutes after each plateau (*+1 Gz* at center of mass), and in the continuous centrifugation group 2 minutes after the plateau was achieved and every 5 minutes thereafter. Mean heart rate, systolic and diastolic blood pressure were calculated for each first measurement intervals during centrifugation on bed rest days 1, 30 and 60 and compared between intervention groups.

Documentation of all adverse events including premature stops, pre-syncopal signs or cardiac dysrhythmias was performed to facilitate evaluation of tolerability.

## Subjective tolerability assessment

General motion sickness susceptibility questionnaire short-form (MSSQ-SF) [29] was determined prior to the head down tilt bed rest including both childhood (MSA) and adulthood (MSB) sub-scores. In both centrifuge groups, Subjective Motion Sickness Ratings (MS: 0 "I am feeling fine" to 20 "I am about to vomit") [30] and rate of perceived exertion (RPE: 6 "No exertion at all" to 20 "Maximal exertion") [31] directly after every centrifuge run during bed rest were recorded. Furthermore, Motion Sickness Assessment Questionnaire (MSAQ), Positive and Negative Affect Schedule (PANAS) and Epworth Sleepiness Scale (ESS) were obtained on a weekly basis directly before, and after centrifugation. MSAQ was employed to determine (1 to 9 max) various dimensions (e.g. gastrointestinal) of motion sickness [32]. PANAS was used to assess the effect of centrifugation upon mood. Participants rated each item on a Likert scale from 1 "not at all" to 5 "very much". The ESS (via rating from 0 (non-) to 3 "high chance of dozing" in 8 contexts) was used to evaluate "drowsiness" since it is a cardinal symptom of motion sickness [33–35]. Furthermore, whenever a centrifuge run was terminated prematurely, the reason was recorded.

## Statistical analysis

Generalized linear mixed models with auto-regressive error AR (1) were used to determine if there was an effect of bed rest (time effect) and intervention (intermittent vs. continuous group). Mean values were reported with standard deviation. All residual plots were evaluated using Kolmogorov-Smirnov with none displaying large deviations from normality. All statistical tests were conducted using IBM SPSS version 21 (IBM Corp., USA) with $\alpha < 0.05$ indicating statistical significance.

## Results

The average spin rate required to generate *+1 Gz* at the center of mass was 30.5 ± 1.0 rpm with radii within 1729–2113 mm at the foot plate. The 16 participants allocated to the two centrifuge groups comprised 10 men and 6 women (71.6 ± 7.4 kg, 33 ± 9.9 yrs, 173 ± 8.8 cm) who experienced 960 centrifuge runs in total.

No serious adverse medical events occurred. However, a total of 10 centrifuge runs (1%, involving 6 different subjects) had to be terminated prematurely; eight runs in the continuous group and two runs in the intermittent group (Fig 2). Of the 10 terminated runs, seven runs–five in the continuous group and two in the intermittent group—had to be terminated due to pre-syncopal signs or symptoms, including significant drop of blood pressure, reporting of tunnel vision and/or lightheadedness. Only one centrifuge run in the continuous group had to be stopped due to severe motion sickness (subsequent MS score of 18/20). Two runs in the continuous group had to be terminated prematurely due to pain resulting from a recent muscle biopsy procedure performed for a different experiment within the bed rest campaign.

No clinically significant cardiac dysrhythmias were observed during centrifugation. During continuous centrifugation, two participants demonstrated frequent isolated premature

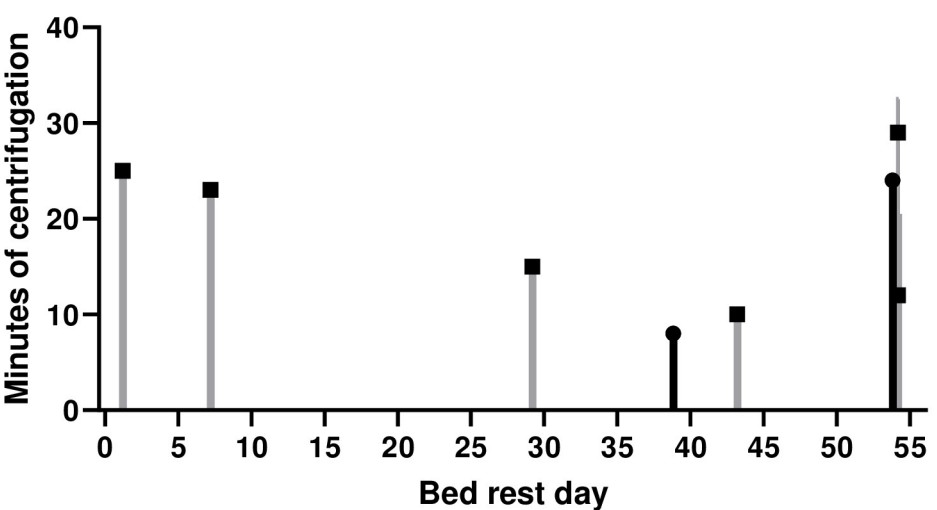

**Fig 2. Premature terminations of centrifuge sessions.**

ventricular complexes on 14 non-consecutive days between bed rest days 5 and 51. Two participants in the continuous group and one in the intermittent centrifugation group exhibited occasional premature atrial complexes, but there was no apparent increase in incidence over time. All subjects were, however, able to resume centrifuge training on subsequent days after a termination.

Comparisons of the initial cardiovascular reactions after 2 minutes of centrifugation on bed rest days 1, 30 and 60 revealed significant effects during bed rest for mean heart rate. Mean heart rates were significantly affected by time for the continuous (F = 14.950, p < 0.001, dfs = 14.073) but not for the intermittent group during bed rest (F = 1.558, p = 0.242, dfs = 15.281). Thus mean heart rate was numerically higher in the continuous group on bed rest day 60 but not significant (continuous group: 100.5 ± 18.5 vs. intermittent group: 86.9 ± 5.9, (t (14) = -1.986, p = 0.67) (Table 1). We observed no significant differences in systolic and diastolic blood pressure.

Overall MSSQ scores were similar (p = 0.211) prior to bed rest with 3.5 ± 5.4 (MSA 1.9 ± 2.7; MSB 1.6 ± 2.7) for the intermittent centrifugation, 6.0 ± 3.7 (MSA 2.9 ± 2.4; MSB 3.1 ± 2.1) for the continuous centrifugation, and 4.7 ± 4.1 (MSA 2.8 ± 2.8; MSB 1.9 ± 3.0) for the control group.

Daily motion sickness scores were significantly higher in the continuous centrifugation group during bed rest (F = 92.8, p = 0.001, dfs = 202.5) with no effect of bed rest time

**Table 1. Comparison of mean values for heart rate, systolic and diastolic blood pressure during the first 2 minutes of centrifugation at the beginning, middle and end of bed rest.**

| | Bed rest phase | | | | | |
|---|---|---|---|---|---|---|
| | **Begin** | | **Middle** | | **End** | |
| | **Intermittent group** | **Continuous group** | **Intermittent group** | **Continuous group** | **Intermittent group** | **Continuous group** |
| **Heart rate** | 80.3 ± 8.4 | 82.4 ± 14.9 | 86.3 ± 12.7 | 99.13 ± 18.6 | 86.9 ± 5.9 | 100.5 ± 18.5 |
| **Systolic blood pressure** | 119.3 ± 13.4 | 111.9 ± 41 | 122.3 ± 10.4 | 128.8 ± 7.5 | 130.5 ± 13.5 | 132.4 ± 11.1 |
| **Diastolic blood pressure** | 76.6 ± 5.9 | 80.8 ± 9.5 | 81.8 ± 4.5 | 84.3 ± 6.3 | 88.1 ± 8.2 | 93.8 ± 11.9 |

## Daily Motion Sickness Rating

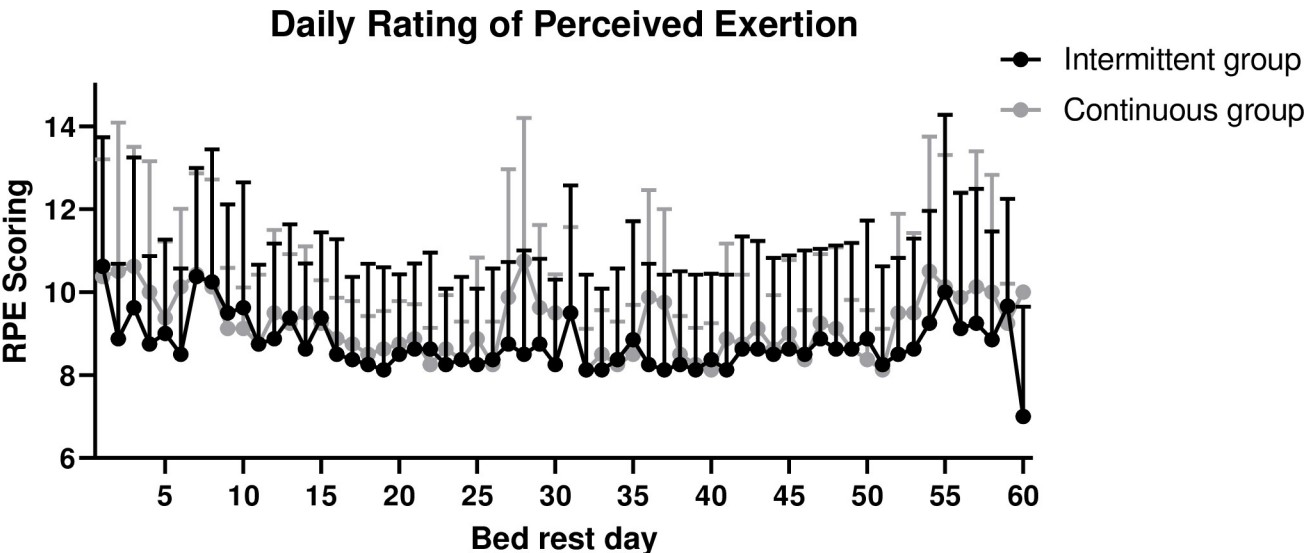

**Fig 3. Mean values with standard error of daily Motion Sickness (MS) rating immediately following intermittent and continuous centrifugation during 60 day bed rest in the intermittent and in the continuous centrifugation group.**

(F = 0.268, p = 0.605, dfs = 217.2) (Fig 3). Pairwise comparison revealed higher motion sickness scores in the continuous (3.05 ± 0.11) compared to the intermittent centrifugation group (1.58 ± 0.11) (p = 0.001).

No significant differences in RPE, MSAQ, PANAS or ESS scores were observed during the bed rest phase neither in either nor between groups (Fig 4, Table 2).

## Discussion

We evaluated the tolerability of daily artificial gravity via short-arm centrifugation as a potential countermeasure against deconditioning induced by 60 day bed rest provided either as a

## Daily Rating of Perceived Exertion

**Fig 4. Mean values with standard error of daily Rating of Perceived Exertion (RPE) rating immediately following intermittent and continuous centrifugation during 60 day bed rest in the intermittent and in the continuous centrifugation group.**

**Table 2. Comparison of tolerability assessment (MSAQ, ESS, PANAS) of both centrifuge intervention groups at the beginning, middle and end of bed rest.**

| | Bed rest phase | | | | | |
|---|---|---|---|---|---|---|
| | Begin | | Middle | | End | |
| | Intermittent group | Continuous group | Intermittent group | Continuous group | Intermittent group | Continuous group |
| **MSAQ** | | | | | | |
| • Overall | 21.96 ± 6.7 | 20.66 ± 3.7 | 16.32 ± 2.2 | 21.76 ± 6.1 | 19.71 ± 4.4 | 17.89 ± 2.5 |
| • Gastrointestinal | 21.18 ± 8.9 | 21.53 ± 4.5 | 12.50 ± 1.0 | 22.22 ± 7.6 | 14.93 ± 3.1 | 18.06 ± 3.4 |
| • Central | 18.06 ± 6.0 | 16.95 ± 3.3 | 13.89 ± 4.1 | 17.52 ± 5.7 | 16.11 ± 6.0 | 14.17 ± 2.5 |
| • Peripheral | 21.30 ± 5.5 | 25.92 ± 8.2 | 15.74 ± 0.6 | 28.39 ± 9.9 | 20.37 ± 3.7 | 18.05 ± 1.8 |
| • Sopite-related | 28.12 ± 7.9 | 20.49 ± 3.9 | 23.61 ± 4.1 | 21.61 ± 3.5 | 28.47 ± 6.4 | 22.22 ± 3.4 |
| **ESS** | 12.5 ± 1.3 | 12.25 ± 1.2 | 15.25 ± 2.3 | 12.56 ± 1.4 | 14.25 ± 2.0 | 13.36 ± 1.8 |
| **PANAS (Positive Affect)** | 23.88 ± 2.6 | 25.75 ± 2.7 | 24.5 ± 2.0 | 24.11 ± 2.0 | 24.50 ± 3.1 | 22.88 ± 3.2 |
| **PANAS (Negative Affect)** | 15.50 ± 1.6 | 13.63 ± 0.5 | 13.13 ± 0.4 | 14.44 ± 1.1 | 15.00 ± 1.2 | 14.25 ± 0.8 |

single 30 min run or as 6x5 minute runs. Our main findings were that both centrifuge interventions were well tolerated (in both males and females), with no serious adverse events and <1% run termination due to pre-syncopal signs. Only a single run was stopped due to motion sickness, with two terminated due to pain from an experimental procedure from another protocol. All subjects were, however, able to resume centrifuge training on subsequent days. Daily motion sickness scores were low, but significantly higher in the continuous group across bed rest. MSAQ, PANAS or ESS scores were low in both centrifugation groups with no difference between groups indicative of good long-term tolerability.

Short-arm centrifugation induces an orthostatic stress on the cardiovascular system that markedly differs from standing on Earth. While the body experiences 1 g terrestrial gravity throughout with standing, the gravitational stimulus increases in a graded fashion from the head towards the feet during short-arm centrifugation [1].

Yet, previous studies have not observed major differences in cardiovascular regulation when standing and during short-arm centrifugation [36]. In our study, pre-syncope occurred in only a few runs and we did not observe overt syncope. Pre-syncope did occur slightly more frequently in the continuous centrifugation group, suggesting that the breaks in the intermittent protocol may contribute to improved orthostatic tolerance during centrifugation. However, in both groups the incidence was very low, potentially due to the fact that subjects were permitted to perform isometric leg muscle pump exercises when experiencing symptoms. In the absence of countermeasures, bed rest deconditioning is associated with markedly reduced orthostatic tolerance [37]. Interestingly, we did not observe worsening tolerability of short-arm centrifugation over time suggesting that daily artificial gravity may have maintained orthostatic tolerance but this requires further evaluation including specific testing of orthostatic tolerance during bed rest [19].

While we did not observe higher degree cardiac dysrhythmias during centrifugation, frequent isolated premature ventricular complexes in two participants in the continuous centrifugation group are noteworthy as long-arm centrifugation nor orthostatic stress imposed by standing are associated with cardiac dysrhythmias in otherwise healthy persons [38, 39]. Whether premature ventricular complexes were triggered by short-arm centrifugation or other stresses resulting from the complex multi-experimental study cannot be discerned. It is reassuring that orthostatic stress imposed by standing or long-arm centrifugation rarely produces significant cardiac dysrhythmias in otherwise healthy persons [38, 39]. While presyncope occurred slightly more frequently in the continuous centrifugation group the incidence is

too small to perform a comprehensive study on intervention group effects. Premature termination of a centrifugation runs were also (albeit rarely) caused by pain due to muscle biopsy from another experiment that were also associated with higher perceived exertion ratings on bed rest days 6 and 55, corroborated by subject comments documented by the attending physician.

As the objective of the present study was to expose subjects to *+1 Gz* at the center of mass and approximately *+2 Gz* at the level of the feet, spin rates during centrifugation were relatively high. During such spin rates head movements can exacerbate motion sickness due to induced conflicts between acceleration (gravity) perception and other sensory inputs [24, 40–42]. However, in our study these spin rates were well tolerated even without physical head restraint or head cover to put subjects into darkness which is commonly used. Remarkably, despite the fact that participants were requested, but not physically prevented from moving the head, only a single centrifuge run was stopped due to severe motion sickness symptoms. Indeed, daily ratings for motion sickness did not indicate increases over time in discomfort due to centrifugation-induced cross-coupled sensations.

Thus, this suggests that by limiting centrifugal acceleration to $5°$ $s^{-2}$ the risk of significant motion sickness is low, even in the intermittent group whom were exposed to multiple acceleration and decelerations within each session. Thus, why higher (albeit not high) motion sickness ratings were reported in the continuous group is unknown and warrants further study–particularly as MSAQ scoring did not differ significantly between groups. Potential limitations of our study are overestimation of questionnaire results as direct comparison with the control group were not obtained due to the complexity of the study. Although our results may be in accordance with other studies showing high levels of vestibular adaption to high speed short radius rotations over time [12, 30, 43] that may also underlie the low scores for PANAS negative affects–suggesting potentially good long-term tolerability.

In conclusion short-arm centrifugation was well tolerated (in both males and females) during 60-days of 6° head-down tilted bedrest. 30 minute intermittent centrifugation appears to be slightly better tolerated compared to equivalent continuous centrifugation indicated by lower motion sickness scores and fewer run terminations. However, the differences were small and require further study in a mixed sex cohort both as 'passive' countermeasures and potentially with concurrent exercise as this may augment effectiveness against multi-systems deconditioning.

## Supporting information

**S1 File. AGBR_quest_results: List of all results from questionnaires ESS, MSAQ and PANAS in a comprehensive manner.**
(XLSX)

**S2 File. Medical data: List of all medical data including heart rate and blood pressure pre and post centrifugation as well as within first two minutes during centrifugation.**
(XLSX)

**S3 File. MS scoring HDT: Recording of motion sickness questionnaires during 60 days of head down tilt bed rest.**
(CSV)

**S4 File. RPE scoring HDT: Recording of perceived exertion questionnaires during 60 days of head down tilt bed rest.**
(CSV)

## Acknowledgments

The authors would like to thank the DLR centrifuge team and DLR study team for their work in the experiment. Further gratitude belongs to NASA and ESA for their cooperation.

## Author Contributions

**Conceptualization:** Timo Frett.

**Data curation:** Timo Frett.

**Formal analysis:** Timo Frett.

**Investigation:** Timo Frett, Michael Arz, Willi Pustowalow, Guido Petrat.

**Methodology:** Timo Frett.

**Project administration:** Timo Frett.

**Supervision:** David Andrew Green, Uwe Tegtbur, Jens Jordan.

**Visualization:** Timo Frett.

**Writing – original draft:** Timo Frett.

**Writing – review & editing:** Timo Frett, David Andrew Green, Edwin Mulder, Alexandra Noppe.

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
