## [Decision Letter · Decision Letter 0]

1 Jul 2020

PONE-D-20-13193

Tolerability and acceptability of daily intermittent or continuous short-arm centrifugation during 60-day 6o head down bed rest

PLOS ONE

Dear Dr. Frett,

Thank you for submitting your manuscript to PLOS ONE. After careful consideration, we feel that it has merit but does not fully meet PLOS ONE’s publication criteria as it currently stands. Therefore, we invite you to submit a revised version of the manuscript that addresses the points raised during the review process.

We look forward to receiving your revised manuscript.

Kind regards,

Shigehiko Ogoh

Academic Editor

PLOS ONE

Journal Requirements:

'The “Artificial Bed Rest study with European Space Agency” (ABRESA) study was financed by NASA and ESA. Our experiments were included into the study conduction and financed by the DLR Institute of Aerospace Medicine.  KBRwyle GmbH provided the salary for D.G but did not have any additional role in the study design, data collection and analysis, decision to publish, or preparation of the manuscript.' 

We note that one or more of the authors are employed by a commercial company: KBRwyle GmbH.

Additional Editor Comments (if provided):

Your manuscript has been reviewed by three experts in the field. They found a merit in this study but they have some major questions. The authors need to address these issues by a careful and considered response.

Reviewers' comments:

Reviewer's Responses to Questions

**Comments to the Author**

1. Is the manuscript technically sound, and do the data support the conclusions?

Reviewer #1: Partly

Reviewer #2: Yes

Reviewer #3: Yes

2. Has the statistical analysis been performed appropriately and rigorously? 

Reviewer #1: I Don't Know

Reviewer #2: Yes

Reviewer #3: Yes

3. Have the authors made all data underlying the findings in their manuscript fully available?

Reviewer #1: Yes

Reviewer #2: Yes

Reviewer #3: Yes

4. Is the manuscript presented in an intelligible fashion and written in standard English?

Reviewer #1: Yes

Reviewer #2: Yes

Reviewer #3: Yes

5. Review Comments to the Author

Reviewer #1: The authors evaluate tolerability and acceptability of intermittent VS continuous short-arm centrifugation during long term head down bed rest. They conclude that both are tolerated, continuous a bit less.

The manuscript is well written but I have a lot of concerns.

First a control group is mentionned but not reported. I would say that before continuous Vs intermittent, the tolerability of daily centrifugation was the main question.

By the way, tolerability /acceptability, can you explain the difference ? Acceptability is not defined in the methods section.

1G was set at the center of mass. Why ? why not HIL, vestibular level, center of bones mass ? how was the center of mass determined ?

Pain is usually evaluated through 10 cm visual analog scale, this a especially important for repeated evaluation to prevent memorisation of the previous score. why did you choose a 0 to 20 scale which could potentially blur longitudinal scoring ?

MSSQ: Your reference is that of Golding that is presumably concerning the MSSQ-short questionnaire, I think you should define MSA and MSB for non-specialists.

At some days it seems that the RPE scoring is higher (see for instance day 55) Can you comment on how other experiments conducted on these subjects (biopsy...) may have influenced tolerability ?

Reviewer #2: PONE-D-20-13193

GENERAL

This report appears to be the first of a number of reports that are based on the AGBRESA study, in which many research groups have studied various aspects of space deconditioning and the effects of artificial gravity as a countermeasure. In order to make these data available in a sound way, too many and too thin “salami” papers should be avoided. This reviewer therefore requests an over-all publication plan (confidential, not to be included in the paper).

The term AGBRESA should be included such that it can be searched (abstract? title?), thereby allowing readers to get the full picture of the AGBRESA results.

SPECIFIC (line numbers introduced by the reviewer- should have been there in the first place)

40: during daily?

104-105 and Fig 1: Is it the area under the curves or the times at top spin rate that are identical between the continuous and the intermittent AG?

110: Radius at the foot plate? (could be computed by the reader from the spin rate, but that should not be necessary)

162-163: Foot pain has been reported in earlier AG studies.

177: p=0.67? tendency?

217: Is ref # 21 really relevant here? Do you mean #19? Missing original rference?

256: Promissory notes regarding future studies are discouraged. Please rephrase.

Reviewer #3: This manuscript addresses an important research question: type of AG to use (that is, intermittent or continuous). I have some suggestions as well as additional references that should be incorporated before this manuscript can be accepted.

1. A lot of text is provided about SANS. You neither measure it nor discuss your results in the context of SANS. I would suggest that you expand the introduction along the lines of the responses that you measured (hemodynamic, vestibular, etc.). Additional references should be placed in. For instance, Iwase et al., 2002; Goswami et al., 2015a; White et al., 2015)

2. Throughout the paper, no mention is made about gender. Then suddenly in the study results, we observe that the study involved males and females. The authors should discuss in the introduction what is the current evidence about sex-based differences in responses in space and/or during AG exposure. See, for example, Evans et al., 2018. Then in the discussion, the results obtained here (sex differences in the responses) should be thoroughly discussed.

3. If you were asked by a researcher, which of these protocols to use for future AG studies, what would you recommend? This could be a clear message of the paper and should be discussed at length in the paper. Currently, it is not clear which AG protocol should be used in the future.

4. Could you also speculate which of these AG protocol would you recommend in the future if additional countermeasures such as nutrition or exercise are also added? See Iwase et al., 2005.

5. Why did some of your participants faint/ develop presyncopal symptoms during AG exposure? Have you tested what their known limit of AG tolerance was? Recent evidence suggests that people take different times to develop presyncope on the centrifuge (Goswami et al.2015b). You should discuss your results in context to this paper as well as come up with clear recommendations about which AG protocol to use based on your evidence as well as published work.

I congratulate the authors for their creativity and thank them for advancing the field of AG.

References in detail

Evans, JM et al. Artificial Gravity as a Countermeasure to the Cardiovascular Deconditioning of Spaceflight: Gender Perspectives. Front Physiol. 2018; 9: 716-716.

Evans, JM et al. Hypovolemic men and women regulate blood pressure differently following exposure to artificial gravity. Eur J Appl Physiol. 2015; 115(12):2631-2640

Goswami et al. Short-arm human centrifugation with 0.4g at eye and 0.75g at heart level provides similar cerebrovascular and cardiovascular responses to standing. Eur J Appl Physiol. 2015a; 115(7):1569-1575

Goswami et al. Effects of individualized centrifugation training on orthostatic tolerance in men and women. PLoS One. 2015b; 10(5):e0125780-e012578

Iwase et al. Effects of Graded Load of Artificial Gravity on Cardiovascular Functions in Humans. Environ Med 2002 Dec;46(1-2):29-32.

White et al.The effects of varying gravito-inertial stressors on grip strength and hemodynamic responses in men and women. Eur J Appl Physiol. 2019; 119(4):951-960

6. PLOS authors have the option to publish the peer review history of their article (what does this mean?). If published, this will include your full peer review and any attached files.

Reviewer #1: **Yes: **Hervé Normand

Reviewer #2: No

Reviewer #3: No

---

## [Author Response · Author response to Decision Letter 0]

27 Jul 2020

Response to Reviewer

Dear Reviewer,

Dear Editor,

Thank you for your helpful comments on our manuscript. We have read them carefully and worked on a revised version that hopefully complies to your requirements. Please find a list of changes based on your input and comments below:

Editor comments:

3. Please include captions for your Supporting Information files at the end of your manuscript, and update any in-text citations to match accordingly.

Answer: 

1.-2.: We have changed all declared parts including funding statement, author’s contribution and competing interests as requested.

3. We have added captions and a short description of all Supporting Information files. 

Reviewer#1 comments:

1. The authors evaluate tolerability and acceptability of intermittent VS continuous short-arm centrifugation during long term head down bed rest. They conclude that both are tolerated, continuous a bit less. The manuscript is well written but I have a lot of concerns. First a control group is mentioned but not reported. I would say that before continuous Vs intermittent, the tolerability of daily centrifugation was the main question. 

Answer: Thank you for pointing this out. Since daily centrifugation has not previously been tested in a longer-term bedrest study, we were concerned that adverse effects, such as syncope, could limit the utility of the intervention with increasing deconditioning and volume changes. If so, the countermeasure would be difficult to conduct during longer term space missions. Thus, from a practical point of view, the main question was how many centrifugation sessions would have to be aborted. Because the control group did not undergo centrifugation, there cannot be an aborted centrifugation. However, some of the more subjective tolerability assessments could be overestimated without adjustment of these measures in the control group. Unfortunately, these measurements were not obtained in the control group due to complexity of the study. We included the issue as potential limitation in the discussion (see lines 261-262). 

2. By the way, tolerability /acceptability, can you explain the difference? Acceptability is not defined in the methods section. 

Answer: We have changed the terminus to tolerability as this is much clearer. 

3. 1G was set at the center of mass. Why? why not HIL, vestibular level, center of bones mass ? How was the center of mass determined ? 

Answer: The centrifuge profile was a priori decided by NASA and ESA together with the DLR centrifuge science team. Previous centrifuge studies such as the BRAG1 study at MEDES (see Clement et al. 2015) showed good tolerability and partially effectiveness of both protocols during 5 days head down tilt bed rest. We have added a brief explanation in line 198-200. The setting in MEDES for BRAG1 was CoM at approximately 1.18 – 1.22 m from the CoR (Center of Rotation). The resulting rotational speed was 27.8 – 28.2, resulting in a +1Gz at CoM and approx. +2Gz at the feet. During AGBRESA we determined the center of mass of the subjects using anthropometry. Rotational speed of the centrifuge was therefore calculated for each subject individually. We also calculated and changed the position of each subject along the centrifuge radius to meet +2Gz at feet accordingly (see lines 112-120).

4. Pain is usually evaluated through 10 cm visual analog scale, this a especially important for repeated evaluation to prevent memorisation of the previous score. why did you choose a 0 to 20 scale which could potentially blur longitudinal scoring ? 

Answer: We recorded motion sickness using a 0 to 20 scale (0 “I am feeling fine” to 20 “I am about to vomit”) that was fast and easy to use during centrifugation according Young et al. 2003. Pain was recorded only by the attending physician when reported by the subject and was not part of our experiment. As reported pain from other experiments (e.g. previous muscle biopsy) caused premature stops of intervention.

5. MSSQ: Your reference is that of Golding that is presumably concerning the MSSQ-short questionnaire, I think you should define MSA and MSB for non-specialists.

Answer: Thank you. We have added a short definition of MSA and MSB in the method section (see lines 138-140)

6. At some days it seems that the RPE scoring is higher (see for instance day 55) Can you comment on how other experiments conducted on these subjects (biopsy...) may have influenced tolerability ?

Answer: We have checked RPE scoring and the documentation of participant’s comments on pain during centrifugation after muscle biopsy experiments and have added this point to the discussion (lines 241-244).

Reviewer#2 comments:

7. This report appears to be the first of a number of reports that are based on the AGBRESA study, in which many research groups have studied various aspects of space deconditioning and the effects of artificial gravity as a countermeasure. In order to make these data available in a sound way, too many and too thin “salami” papers should be avoided. This reviewer therefore requests an over-all publication plan (confidential, not to be included in the paper).

Answer: ESA and NASA selected principal investigators for the AGBRESA study using a peer review approach. Each principal investigator has the right to publish experiments conducted as part of the AGBRESA study, however, there are some differences between NASA and ESA regulations. There is no overall-publication plan. 

8. The term AGBRESA should be included such that it can be searched (abstract? title?), thereby allowing readers to get the full picture of the AGBRESA results.

Answer: In accordance with the comment above we have included the term AGBRESA also in the title.

9. SPECIFIC (line numbers introduced by the reviewer- should have been there in the first place)

40: during daily?

Answer: Corrected, thank you.

104-105 and Fig 1: Is it the area under the curves or the times at top spin rate that are identical between the continuous and the intermittent AG?

Answer: Due to the study design the times at top spin rate were identical, so both groups were exposed to daily 30 minutes AG with +1Gz at CoM. The intermittent group had a longer exposure to AG due to the frequent ramp up/down. 

110: Radius at the foot plate? (could be computed by the reader from the spin rate, but that should not be necessary)

Answer: We calculated an individual centrifuge speed and position along centrifuge radius to ensure +1Gz at CoM and +2Gz at the foot plate. These radii were within 1729-2113 mm (see lines 112-120). E. Mulder et al. will publish a more general method paper elsewhere to give further details. 

162-163: Foot pain has been reported in earlier AG studies.

177: p=0.67? tendency?

Answer: We changed the sentences to be more precise. 

217: Is ref # 21 really relevant here? Do you mean #19? Missing original reference?

Answer: You are right, we changed the relevant reference.

256: Promissory notes regarding future studies are discouraged. Please rephrase.

Answer: As suggested, we rewrote the section. 

Reviewer #3 comments: 

1. A lot of text is provided about SANS. You neither measure it nor discuss your results in the context of SANS. I would suggest that you expand the introduction along the lines of the responses that you measured (hemodynamic, vestibular, etc.). Additional references should be placed in. For instance, Iwase et al., 2002; Goswami et al., 2015a; White et al., 2015)

Answer: Thank you. We have added missing sentences and references in the introduction.

2. Throughout the paper, no mention is made about gender. Then suddenly in the study results, we observe that the study involved males and females. The authors should discuss in the introduction what is the current evidence about sex-based differences in responses in space and/or during AG exposure. See, for example, Evans et al., 2018. Then in the discussion, the results obtained here (sex differences in the responses) should be thoroughly discussed.

Answer: We reported gender distribution of the study in the method section (see line 101) and have added recent findings from Evans et al. 2018 in the introduction. We discussed internally a sex based analysis of our results but the limited sample size (n = 8 per group) do not allow well powered statistics about gender differences. This topic was therefore also discussed with ESA. For the two upcoming bed rest studies involving AG as intervention will use two groups with n = 12 participants each. We also concluded in the discussion that a mixed sex cohort is required to further investigate AG as potential countermeasure. 

3. If you were asked by a researcher, which of these protocols to use for future AG studies, what would you recommend? This could be a clear message of the paper and should be discussed at length in the paper. Currently, it is not clear which AG protocol should be used in the future.

Answer: You are right. We have rewritten and added additional text in the discussion section. 

4. Could you also speculate which of these AG protocol would you recommend in the future if additional countermeasures such as nutrition or exercise are also added? See Iwase et al., 2005.

Answer: We have rewritten the discussion to make the planned combination of AG+exercises more clear. 

5. Why did some of your participants faint/ develop presyncopal symptoms during AG exposure? Have you tested what their known limit of AG tolerance was? Recent evidence suggests that people take different times to develop presyncope on the centrifuge (Goswami et al.2015b). You should discuss your results in context to this paper as well as come up with clear recommendations about which AG protocol to use based on your evidence as well as published work.

Answer: During the Baseline data collection phase of AGBRESA only two short familiarization sessions with centrifugation were allowed for each subject. Therefore individual AG tolerance testing as shown by Goswami et al. was unfortunately not allowed by the study setting. To the authors knowledge this topic is currently under discussion by the ESA bed rest expert team for both upcoming bed rest studies in MEDES and Planica.

---

## [Decision Letter · Decision Letter 1]

19 Aug 2020

PONE-D-20-13193R1

Tolerability of daily intermittent or continuous short-arm centrifugation during 60-day 6o head down bed rest (AGBRESA study)

PLOS ONE

Dear Dr. Frett,

Thank you for submitting your manuscript to PLOS ONE. After careful consideration, we feel that it has merit but does not fully meet PLOS ONE’s publication criteria as it currently stands. Therefore, we invite you to submit a revised version of the manuscript that addresses the points raised during the review process.

We look forward to receiving your revised manuscript.

Kind regards,

Shigehiko Ogoh

Academic Editor

PLOS ONE

Reviewers' comments:

Reviewer's Responses to Questions

**Comments to the Author**

1. If the authors have adequately addressed your comments raised in a previous round of review and you feel that this manuscript is now acceptable for publication, you may indicate that here to bypass the “Comments to the Author” section, enter your conflict of interest statement in the “Confidential to Editor” section, and submit your "Accept" recommendation.

Reviewer #2: All comments have been addressed

Reviewer #3: All comments have been addressed

2. Is the manuscript technically sound, and do the data support the conclusions?

Reviewer #2: Yes

Reviewer #3: Yes

3. Has the statistical analysis been performed appropriately and rigorously? 

Reviewer #2: I Don't Know

Reviewer #3: Yes

4. Have the authors made all data underlying the findings in their manuscript fully available?

Reviewer #2: Yes

Reviewer #3: Yes

5. Is the manuscript presented in an intelligible fashion and written in standard English?

Reviewer #2: Yes

Reviewer #3: Yes

6. Review Comments to the Author

Reviewer #2: PONE-D-20-13193R1

GENERAL

My comments and questions have been adequately addressed. With respect to a publication plan I note that a method paper by Mulder is planned. Please consider merging that with the present ms for a more solid paper.

SPECIFIC

Line 153: “Statistical analysis. Generalized linear mixed models with auto-regressive error

AR(1) were used to determine if there was an effect of bed rest (time effect) and intervention (intermittent vs. continuous group). Mean values were reported with standard

Deviation”.

This reviewer is not sure that the above statistical approach is appropriate for rated estimates of motion sickness. Please consult an expert. Sorry for not pointing this out in my previous review.

Reviewer #3: Most of my comments have been addressed.

However, reference numbers 21 and 23 are the same.

Reference number 23 should be replaced by: Goswami, N; Blaber, AP; Hinghofer-Szalkay, H; Convertino, VA. Lower Body Negative Pressure: Physiological Effects, Applications, and Implementation. Physiol Rev. 2019; 99(1): 807-851

7. PLOS authors have the option to publish the peer review history of their article (what does this mean?). If published, this will include your full peer review and any attached files.

Reviewer #2: No

Reviewer #3: No

---

## [Author Response · Author response to Decision Letter 1]

24 Aug 2020

Response to Reviewer

Dear Ladies and Gentlemen,

We thank the editor and both reviewers for helpful comments on our manuscript. We reviewed the comments carefully and revised the manuscript accordingly. Please find a list of changes based on your input and comments below:

Reviewer#2 comments:

1. GENERAL

My comments and questions have been adequately addressed. With respect to a publication plan I note that a method paper by Mulder is planned. Please consider merging that with the present ms for a more solid paper.

Answer: We have carefully discussed your suggestion. A method paper about AGBRESA is in preparation. However, the manuscript is rather dense and contains very detailed information regarding study design, including recruitment and methods for standardization, demographics, vital signs, and dietary composition among others. All other data including the bed rest core data will be embedded in manuscripts by principial investigators of individual experiments funded by DLR, ESA or NASA. Given the potential application of artificial gravity as countermeasure in space and a series of future studies testing this modality in combination with exercise, we believe that a detailed analysis of tolerability should be made available to the scientific community. 

2. SPECIFIC

Line 153: “Statistical analysis. Generalized linear mixed models with auto-regressive error AR(1) were used to determine if there was an effect of bed rest (time effect) and intervention (intermittent vs. continuous group). Mean values were reported with standard Deviation”. This reviewer is not sure that the above statistical approach is appropriate for rated estimates of motion sickness. Please consult an expert. Sorry for not pointing this out in my previous review.

Answer: Thank you for pointing this out. We have consulted an experienced biometrician prior to the data analysis and have discussed the correct method to analyze the data. The biometrician advised us to use this linear mixed model. 

Reviewer #3 comments:

1. Most of my comments have been addressed. However, reference numbers 21 and 23 are the same. Reference number 23 should be replaced by: Goswami, N; Blaber, AP; Hinghofer-Szalkay, H; Convertino, VA. Lower Body Negative Pressure: Physiological Effects, Applications, and Implementation. Physiol Rev. 2019; 99(1): 807-851

Answer: Thank you for this note, we have corrected the error and included the additional reference.

---

## [Editor Report · Decision Letter 2]

2 Sep 2020

Tolerability of daily intermittent or continuous short-arm centrifugation during 60-day 6o head down bed rest (AGBRESA study)

PONE-D-20-13193R2

Dear Dr. Frett,

We’re pleased to inform you that your manuscript has been judged scientifically suitable for publication and will be formally accepted for publication once it meets all outstanding technical requirements.

Kind regards,

Shigehiko Ogoh

Academic Editor

PLOS ONE

Additional Editor Comments (optional):

Thank you for your work.　No further comment.
---

## [Editor Report · Acceptance letter]

10 Sep 2020

PONE-D-20-13193R2 

Tolerability of daily intermittent or continuous short-arm centrifugation during 60-day 6^o^ head down bed rest (AGBRESA study) 

Dear Dr. Frett:

I'm pleased to inform you that your manuscript has been deemed suitable for publication in PLOS ONE. Congratulations! Your manuscript is now with our production department. 

Kind regards, 

on behalf of

Dr. Shigehiko Ogoh 

Academic Editor

PLOS ONE